



# Technical note: Sequential ensemble data assimilation in convergent and divergent systems

Hannes Helmut Bauser[1,2], Daniel Berg[2,3], and Kurt Roth[2,4]

[1]Biosphere 2, University of Arizona, USA
[2]Institute for Environmental Physics (IUP), Heidelberg University, Germany
[3]HGS MathComp, Heidelberg University, Germany
[4]Interdisciplinary Center for Scientific Computing (IWR), Heidelberg University, Germany

**Correspondence:** Hannes Helmut Bauser (hbauser@arizona.edu)

**Abstract.** Data assimilation methods are used throughout the geosciences to combine information from uncertain models and uncertain measurement data. However, the characteristics of geophysical systems differ and may be distinguished between divergent and convergent systems. In divergent systems initially nearby states will drift apart, while they will coalesce in convergent systems. This difference has implications for the application of sequential ensemble data assimilation methods. This study explores these implications on two exemplary systems: the divergent Lorenz-96 model and the convergent description of soil water movement by the Richards equation. The results show that sequential ensemble data assimilation methods require a sufficient divergent component. This makes the transfer of the methods from divergent to convergent systems challenging. We demonstrate through a set of case studies that it is imperative to represent model errors adequately and incorporate parameter uncertainties in ensemble data assimilation in convergent systems.

## 1 Introduction

Information on physical systems is often available in two forms: on the one hand from observations and on the other hand through mathematical models describing the systems dynamics. The combination of both can lead to an improved description of the system. This is the aim of data assimilation, typically with a focus on state estimation.

Data assimilation has broad applications throughout the geosciences and can be already seen as an independent discipline (Carrassi et al., 2018). It is typically used to estimate states, but also parameters: in weather forecasting (Houtekamer and Zhang, 2016; Ruiz et al., 2013), for atmospheric chemical transport (Carmichael et al., 2008; Zhang et al., 2012) also coupled to meteorology (Bocquet et al., 2015), in oceanography including biogeochemical processes (Stammer et al., 2016; Edwards et al., 2015), and in hydrology for flow, transport, and reaction in terrestrial surface and subsurface systems (Liu et al., 2012). Data assimilation is also increasingly applied in ecology with applications ranging from the spread of infectious diseases and wildfires, to population dynamics, and to the terrestrial carbon cycle (Niu et al., 2014; Luo et al., 2011).

The geophysical systems may be distinguished between divergent and convergent systems, depending on the development of two initially nearby states. In a divergent system, initially close states will inevitably drift apart, even if the system is described by a perfect model (Kalnay, 2003). This leads to an upper limit for the predictability in divergent systems (Lorenz, 1982). In a





convergent system, nearby trajectories will coalesce. If the model to describe such a convergent system is perfect, this results
in a high predictability (Lorenz, 1996). An error in the initial state will decay towards the truth after some transient phase.
However, this is only true for perfect models, which is usually not the case for geophysical systems. This can lead to a bias
with the state converging to a wrong state.

Many recent advances of data assimilation methods have been developed in the context of weather forecasting (Van Leeuwen
et al., 2015). They are therefore designed to meet the challenges in the atmosphere – a divergent system. Due to the fundamental
limit for long time predictions from uncertain initial conditions in divergent systems, data assimilation in operational weather
forecasting primarily focuses on state estimation (Reichle, 2008; Van Leeuwen et al., 2015).

Several geophysical system, such as soil hydrology or atmospheric chemical transport are convergent systems. In ensemble
data assimilation methods, where uncertainties are represented through an ensemble of states, this leads to decreasing uncer-
tainties over time, which favors filter degeneration. This difference to divergent systems, where the ensemble spread increases
exponentially, makes the direct transfer of data assimilation methods from divergent systems to convergent systems challenging
and often requires adaptations to prevent filter degeneracy.

The largest uncertainties in convergent systems typically do not reside in uncertain initial conditions but rather in boundary
conditions, the representation of sub-scale physics through parameterizations, and unrepresented physics in the model equa-
tions. These uncertainties should then be addressed integratedly (Liu and Gupta, 2007). Therefore, data assimilation methods
have been used to not only estimate states but to also estimate parameters to reduce these uncertainties. The combined esti-
mation of states and parameters is thought to be a solution of reducing the impact of model errors on parameter estimation
(Liu et al., 2012). Estimating parameters in ensemble data assimilation methods through an augmented state requires a forward
model for the parameters as well. This model is typically assumed to be constant, which is neither divergent nor convergent.
However, the filter will gradually reduce the uncertainty in the parameters, which is not increased through a divergent forward
model and challenges similar to convergent systems can arise.

A challenge for sequential ensemble data assimilation in convergent systems it to maintain a sufficient ensemble spread. One
possibility are inflation methods, which counteract the coalescing tendency. Unfortunately, there exist no universal method to
accomplish this and a range of approaches are followed. One example is the increase of the ensemble spread of parameters
to a threshold value, as soon as the parameter uncertainty drops below this value. This approach has was introduced for a
see-breeze model (Aksoy et al., 2006) and has been used in hydrology, as well (e.g. Shi et al., 2014; Rasmussen et al., 2015). A
modification to this, also applied in hydrology, is to keep the parameter uncertainty entirely constant (Han et al., 2014; Zhang
et al., 2017). This ensures a sufficient ensemble spread in the state itself, but can impact the accuracy of the estimation. A
widely used adjustment to limit the reduction of an ensemble spread in hydrology is the use of a damping factor (Franssen and
Kinzelbach, 2008). Also in hydrology, a multiplicative inflation method was proposed, specifically adjusted to the needs of
the system (Bauser et al., 2018). Similarly, Constantinescu et al. (2007) showed that an atmospheric chemical transport model
required stronger inflation and showed better results for a model specific inflation, where the key parameters are perturbed to
achieve an increased spread in the state. Consequently, a better understanding and control over errors has been recognized as a
major challenge in chemical data assimilation as well (Zhang et al., 2012).





Although plenty of knowledge and experience is available in the different communities how to handle data assimilation
methods in their specific model, we are not aware of a fundamental analysis of the difference between divergent and convergent
models with respect to their utilization within ensemble data assimilation frameworks. We investigate and demonstrate the
different challenges using the ensemble Kalman filter (Evensen, 1994; Burgers et al., 1998). The divergent case is illustrated
using the Lorenz-96 model (Lorenz, 1996), while for the convergent case, a soil hydrological system described by Richards
equation is used.

## 2   Data assimilation method

For this study we chose the ensemble Kalman filter (EnKF), a sequential data assimilation method, based on Bayes' theorem
and the assumption of unbiased Gaussian error distributions. It was introduced as an extension of the Kalman filter (Kalman,
1960) for nonlinear models (Evensen, 1994; Burgers et al., 1998) and approximates the Gaussian distributions by an ensemble
of states $\psi_i$, where the subscript $i$ denotes the ensemble member.

To sequentially assimilate new observations, the EnKF alternates between a forecast (subscript 'f') and an analysis step
(subscript 'a'). In the forecast, the ensemble is propagated using the nonlinear model $f(\cdot)$

$$\psi_{\mathrm{f},i}^k = f(\psi_{\mathrm{f},i}^{k-1}) + \boldsymbol{\beta}^k \,, \tag{1}$$

where $\boldsymbol{\beta}^k$ is a stochastic model error and $k$ is a discrete time.

The analysis state $\psi_{\mathrm{a}}$ is calculated by applying the Kalman gain

$$\mathbf{K}^k = \mathbf{P}_{\mathrm{f}}^k \mathbf{H}^{\mathsf{T}} (\mathbf{H}\mathbf{P}_{\mathrm{f}}^k \mathbf{H}^{\mathsf{T}} + \mathbf{R}^k)^{-1} \tag{2}$$

to every ensemble member. The Kalman gain weights the forecast error covariance $\mathbf{P}$ with the observation error covariance $\mathbf{R}$.
The observation operator $\mathbf{H}$ maps the state from state space to observation space. The forecast error covariance $\mathbf{P}$ is calculated
using the forecast ensemble

$$\mathbf{P}_{\mathrm{f}}^k = \overline{\left(\psi_{\mathrm{f}}^k - \overline{\psi_{\mathrm{f}}^k}\right)\left(\psi_{\mathrm{f}}^k - \overline{\psi_{\mathrm{f}}^k}\right)^{\mathsf{T}}} \,. \tag{3}$$

It is necessary to add a realization of the observation error to the observation for each ensemble member (Burgers et al.,
1998). The resulting analysis state is

$$\psi_{\mathrm{a},i}^k = \psi_{\mathrm{f},i}^k + \mathbf{K}^k \left(\boldsymbol{d}^k - \mathbf{H}\psi_{\mathrm{f},i}^k + \boldsymbol{\epsilon}_i^k\right) \,, \quad \epsilon_i^k \propto \mathcal{N}(\mathbf{0}, \mathbf{R}^k) \,. \tag{4}$$

For sequential data assimilation, the process of forecast and analysis is iterated for every new observation.

The joint estimation of states and parameters can be realized through state augmentation. The original state $\psi$ is extended
with the parameters $\boldsymbol{p}$ to an augmented state

$$\boldsymbol{u} = \begin{pmatrix} \psi \\ \boldsymbol{p} \end{pmatrix} \,. \tag{5}$$





The model equation (Eq. 1) changes to

$$
\boldsymbol{u}^k = \begin{pmatrix} \boldsymbol{\psi}^k \\ \boldsymbol{p}^k \end{pmatrix} = \begin{pmatrix} f_{\boldsymbol{\psi}}\left(\boldsymbol{\psi}^{k-1}, \boldsymbol{p}^{k-1}\right) + \boldsymbol{\beta}_{\boldsymbol{\psi}}^k \\ f_{\boldsymbol{p}}\left(\boldsymbol{p}^{k-1}\right) + \boldsymbol{\beta}_{\boldsymbol{p}} \end{pmatrix} , \tag{6}
$$

with the models for the state $f_{\boldsymbol{\psi}}(\cdot)$ and for the parameters $f_{\boldsymbol{p}}(\cdot)$, as well as the corresponding model errors $\boldsymbol{\beta}_{\boldsymbol{\psi}}$ and $\boldsymbol{\beta}_{\boldsymbol{p}}$. The model for the state is typical nonlinear, while the parameters are often assumed constant $f_{\boldsymbol{p}}(\boldsymbol{p}^{k-1}) = \boldsymbol{p}^{k-1}$.

In this study, we set both stochastic model errors in Eq. (6) to zero. The EnKF is used without any extensions and an ensemble size of $N = 100$ was chosen for all cases.

## 3   Divergent system

This section demonstrates the data assimilation behavior for a divergent system on the example of the 40-dimensional Lorenz-96 model. We first introduce the model before we look into four characteristic different cases.

### 3.1   Lorenz-96

The Lorenz-96 model (Lorenz, 1996) is an artificial model and cannot be derived from any dynamic equation (Lorenz, 2005). It can be interpreted as an unspecified scalar quantity $x$ in a one dimensional atmosphere on a latitude circle and was defined in a study on predictability (Lorenz, 1996).

The governing equations are a set of coupled ordinary differential equations:

$$
\frac{dx_i}{dt} = (x_{i+1} - x_{i-2})x_{i-1} - x_i + F \quad , \quad i \in [1, 2, \ldots, J] \tag{7}
$$

with constant forcing $F$, periodic boundaries ($x_{J+1} = x_1$) and dimension $J$. The dimension is often chosen as $J = 40$, which we also do in this study. Even though the system is not derived from physical principles, it shares certain properties of large atmospheric models (Lorenz and Emanuel, 1998). The quadratic terms represent advection and conserve the total energy, while the linear term decreases the total energy comparable to dissipation. The constant $F$ represents external forcing and prevents the system's total energy from decaying to zero. The value is often chosen as $F = 8$ (Lorenz and Emanuel, 1998).

The Lyapunov exponent quantifies the local speed of separation of two infinitesimal close trajectories. Analyzing the averaged leading Lyapunov exponent for $F = 8$ shows a doubling time of 0.42 units for the distance between two neighboring states (Lorenz and Emanuel, 1998). However, the growth rate in a short interval can be considerably larger or smaller. A local linear stability analysis shows that the Lorenz-96 model has 13 positive Lyapunov exponents (Lorenz and Emanuel, 1998). This means that 13 directions of the eigenbasis are divergent, while the other 27 directions are convergent.

Increasing the forcing leads to a more divergent system. For $F = 10$ the Lorenz-96 model has 14 positive Lyapunov exponents (instead of 13). Furthermore, the doubling time, decreases from 0.42 to 0.3 (Lorenz, 1996), which also decreases the predictability of the system.





## 3.2 Characteristic cases

The model is solved using a fourth order Runge-Kutta method with a time step of $\Delta t = 0.01$. The behavior of the EnKF on a divergent system is investigated through four different cases (DC1-4).

The initial condition for the synthetic truth for all cases is generated by running the model until time 2000 from an initial state $x_i = 4.0 \, \forall i \in [1, 2, \ldots, 39]$ and $x_{40} = 4.001$, with the typical value $F = 8$ for the forcing parameter. The final state of this run is used as the synthetic true initial state for all cases. This ensures that the state is on the attractor without the initial transient phase. The initial ensemble for the data assimilation runs is generated by perturbing the true initial state with a Gaussian distribution $\mathcal{N}(0, 1)$.

Synthetic observations are generated in all 40 dimensions by a forward run until time 4, using the true value and perturbing it with a Gaussian distribution with a zero mean and a standard deviation of $\sigma_{\mathrm{obs}} = 1.0$. For cases DC1 and DC2 observations are generated at 8 different times with an observation interval of $\Delta t_{\mathrm{Obs}} = 0.5$. This observation interval is chosen rather large to ensure a large divergence of the system. For cases DC3 and DC4 observations are generated at 80 different times with an observation interval of $\Delta t_{\mathrm{Obs}} = 0.05$. This interval is often used in other studies (e.g. Nakano et al., 2007; van Leeuwen, 2010; Poterjoy, 2016).

### 3.2.1 Divergent case 1 (DC1) – state estimation, true parameter

In this case, the ensemble is propagated with the same parameter ($F = 8$) as the synthetic truth. Figure 1a shows the time development of one state dimension of the Lorenz-96 model (top panel) and the ensemble variance of this state dimension $\sigma_{\mathrm{dim}}^2$ together with the mean variance over all dimensions $\sigma^2$ (bottom panel). The ensemble has a sufficient ensemble spread such that the EnKF is able to correct the states to follow the truth.

The mean ensemble spread $\sigma^2$ increases exponentially between two observations. At each observation, the EnKF updates the ensemble and the variance decreases correspondingly.

The behavior of $\sigma_{\mathrm{dim}}^2$ differs from the behavior of $\sigma^2$. During the forecast, $\sigma_{\mathrm{dim}}^2$ sometimes increases, decreases or stays approximately constant. This occurs because the Lorenz-96 model is bounded and has, therefore, convergent and divergent directions.

### 3.2.2 Divergent case 2 (DC2) – state estimation, wrong parameter

In this case, the ensemble is propagated with a wrong parameter of $F = 10$ instead of $F = 8$ (Fig. 1b), which was used to generate the observations. Therefore, the ensemble is propagated with a model that is more divergent than the synthetic truth.

Compared to DC1 (Fig. 1a, bottom panel), $\sigma^2$ and $\sigma_{\mathrm{dim}}^2$ increase faster and the ensemble spread reaches higher values, which shows the increased divergence of the system. Propagating the ensemble with this different model leads to a larger deviation of the ensemble mean from the truth (see Fig. 1b, top panel) than in DC1. However, the divergent nature of the model ensures a sufficient ensemble spread such that the state can be corrected towards the truth and the the wrong parameter is compensated. This leads to a worsened but still good estimation of the truth without filter divergence.



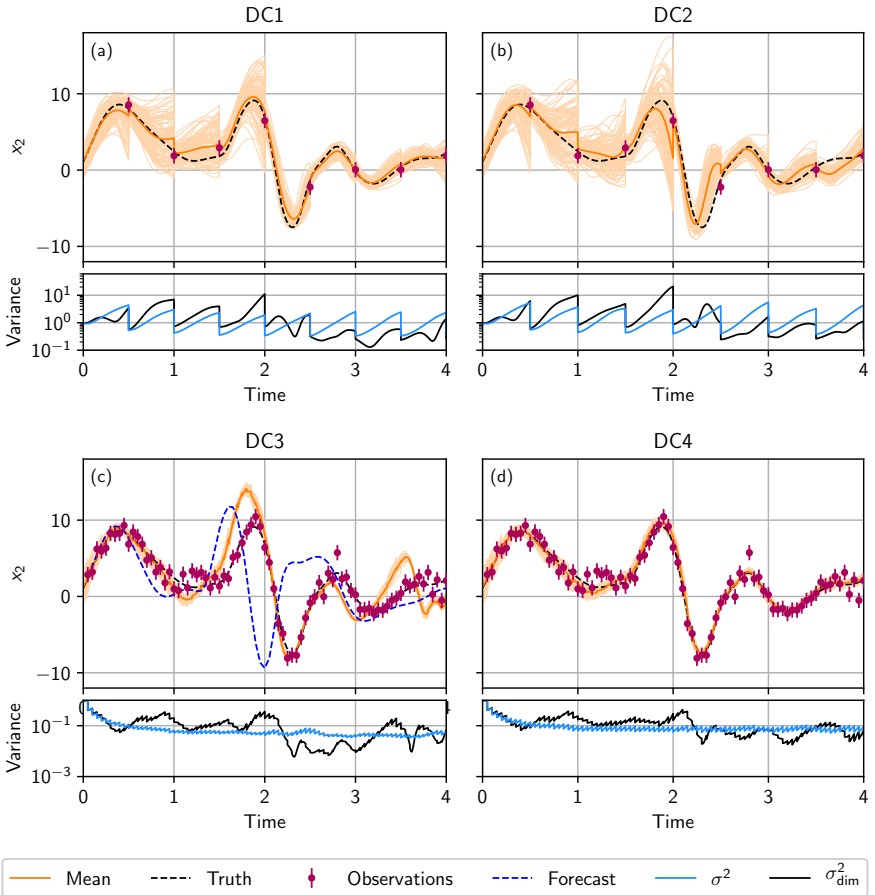

**Figure 1.** State estimation in a divergent system. (a): Divergent case 1 (DC1), the ensemble is propagated with the same parameter as the truth ($F = 8$). (b): Divergent case 2 (DC2), the ensemble is propagated with $F = 10$ instead. (c): Divergent case 3 (DC3), the ensemble is propagated with $F = 10$ and the observation interval is reduced to $\Delta t_{\mathrm{Obs}} = 0.05$. (d): Divergent case 4 (DC4), the parameter error is represented by the ensemble using $\mathcal{N}(10, 2^2)$ for the reduced observation interval of $\Delta t_{\mathrm{Obs}} = 0.05$. Top panels : The ensemble mean (orange line) and the ensemble (light orange lines) in the data assimilation run for state dimension 2 together with the observations (purple), generated from the truth (black dashed line). (c) additionally shows a single forward run (blue dashed line) using a wrong parameter $F = 10$ starting from the true initial condition. Bottom panels: Mean variance $\sigma^2$ (light blue line) of the ensemble over all dimensions and variance $\sigma^2_{\mathrm{dim}}$ (black line) of state dimension 2.

### 3.2.3 Divergent case 3 (DC3) – state estimation, small observation interval, wrong parameter

The case is similar to DC2 except that the observation interval is reduced to $\Delta t_{\mathrm{Obs}} = 0.05$. The high frequency of analysis steps, where the Kalman update reduces the variance in the ensemble, prevents the Lorenz-96 model to develop enough divergence

150 to increase the ensemble spread sufficiently (see Fig. 1a, bottom panel). Although the ensemble is propagated with a more divergent model ($F = 10$), the divergence is not sufficient to encompass the model error due to the wrong parameter and the





filter can degenerate (see Fig. 1c, top panel). This is in contrast to DC2, where the Kalman filter can successfully estimate the state. However, the comparison with the forecast of the true initial state using the wrong parameter shows that the EnKF is able to improve the state significantly.

### 155    3.2.4    Divergent case 4 (DC4) – state estimation, small observation interval, represented error

In divergent case 4 (DC4), the observation interval is, as in DC3, reduced to $\Delta t_{\text{Obs}} = 0.05$. The parameter error is represented by the ensemble by assigning each ensemble member a different parameter $F$. The forcing parameter $F$ is drawn from a Gaussian distribution $\mathcal{N}(10, 2^2)$, such that the true value lies within one standard deviation .

Representing the error increases $\sigma^2$ only slightly (Fig. 1d, bottom panel) compared to DC3, but the minimal variance of $\sigma^2_{\text{dim}}$
160   has higher values. This change is sufficient to help the filter such that it does not degenerate (Fig. 1d, top panel). Representing the parameter error in the case of frequent observations can prevent filter degeneracy.

## 4    Convergent system

This section demonstrates the data assimilation behavior for a convergent system on the example of soil water flow. We again first introduce the model before we look into four characteristic cases.

### 165    4.1    Soil water flow

Water flow in an unsaturated porous medium can be described by the Richards equation:

$$\partial_t \theta - \nabla \cdot \left[ K(\theta) \left[ \nabla h_m(\theta) - 1 \right] \right] = 0 , \tag{8}$$

where $\theta$ $(-)$ is the volumetric water content, $K$ $(\mathrm{L T^{-1}})$ is the isotropic hydraulic conductivity, and $h_m$ $(\mathrm{L})$ is the matric head.

To close Eq. 8 soil hydraulic material properties are necessary, which specify the dependency of the matric head and the hy-
170   draulic conductivity on the water content. We use the Mualem-van Genuchten parametrisation (Mualem, 1976; Van Genuchten, 1980) in its simplified form:

$$K(\Theta) = K_w \, \Theta^\tau \left[ 1 - \left[ 1 - \Theta^{n/[n-1]} \right]^{1-1/n} \right]^2 , \tag{9}$$

$$h_m(\Theta) = \frac{1}{\alpha} \left[ \Theta^{-n/[n-1]} - 1 \right]^{1-1/n} , \tag{10}$$

with the saturation $\Theta$ $(-)$

175   $$\Theta := \frac{\theta - \theta_r}{\theta_s - \theta_r} . \tag{11}$$

The parameter $\theta_s$ $(-)$ is the saturated water content and $\theta_r$ $(-)$ is the residual water content. The matric head $h_m$ is scaled with the parameter $\alpha$ $(\mathrm{L^{-1}})$ that can be related to the inverse air entry value. The parameter $K_w$ $(\mathrm{L T^{-1}})$ is the saturated hydraulic





conductivity, $\tau\ (-)$ a tortuosity factor and $n\ (-)$ is a shape parameter. Equation (9) and Eq. (10) describe the sub-scale physics with six parameters for a homogeneous soil.

## 4.2 Characteristic cases

The behavior of the EnKF on a convergent system is investigated through four different cases (CC1-4). For the case studies, a one-dimensional homogeneous soil is used with an extent of $1\,\mathrm{m}$. The Richards equation is solved using MuPhi (Ippisch et al., 2006) with a spatial resolution of $1\,\mathrm{cm}$, which results in a 100-dimensional water content state.

For the true trajectories and the observations, parameters for a loamy sand by Carsel and Parrish (1988) are used: $\theta_s = 0.41$, $\theta_r = 0.057$, $\tau = 0.5$, $n = 2.28$, $\alpha = -12.4\,\mathrm{m}^{-1}$, and $K_w = 4.00 \cdot 10^{-5}\,\mathrm{m\,s}^{-1}$. For the lower boundary, a Dirichlet condition with zero potential (groundwater table) is set and for the upper boundary a constant infiltration over the whole observation time with a flux of $5 \cdot 10^{-7}\,\mathrm{m\,s}^{-1}$ is used.

Initially, the system is in hydraulic equilibrium. The infiltration boundary condition leads to a downward propagating infiltration front increasing the water content. Four water content observations are generated equidistantly at depths of $(0.2, 0.4, 0.6, 0.8)$ m. The observation error is chosen to be $\sigma_{\mathrm{Obs}} = 0.007$ (e.g., Jaumann and Roth, 2017). Observations are taken hourly for a duration of $30\,\mathrm{h}$.

To generate the initial ensemble, the ensemble mean is perturbed by a correlated multivariate Gaussian distribution. The main diagonal of the covariance matrix is $0.003^2$. The off-diagonal entries are determined by multiplying the variance on the main diagonal with the Gaspari and Cohn function (Gaspari and Cohn, 1999) using a length scale of $c = 10\,\mathrm{cm}$. This ensures a correlated initial state, which increases the diversity of the ensemble. If instead uncorrelated Gaussian random numbers with a zero mean were used, the dissipative component of the system would lead to a fast dissipation of the perturbation.

### 4.2.1 Convergent case 1 (CC1) – no estimation

In this case no data assimilation is used and the ensemble is propagated with the true model. The initial conditions for the ensemble members are based on an on the linearly interpolated observations at time zero. This approximated state is used as the ensemble mean for the EnKF. This state is then perturbed by a correlated multivariate Gaussian distribution such that the spread of the initial ensemble is sufficient to represent the uncertainty of the water content in most parts.

The temporal development of the water content at $20\,\mathrm{cm}$ depth, the position of the uppermost observation, is shown in Fig. 2a (top panel). The initially broad ensemble slowly collapses to the truth.

The ensemble variance in this depth $\sigma_{\mathrm{dim}}^2$ increases when the infiltration front reaches $20\,\mathrm{cm}$. Because of the nonlinear conductivity function (Fig. 9), the different initial water contents lead to a different arrival time of the infiltration front at the first observation position. This leads to an increase in the ensemble spread.

After the increase of the water content, the ensemble collapses fast since the hydraulic conductivity increases with the water content, which leads to a fast convergence of the different ensemble members to the truth due the convergent nature of the model. The variance over all dimensions $\sigma^2$ decreases slowly and approaches zero over time. If the system is started with a higher water content instead of equilibrium, this collapse will occur faster.



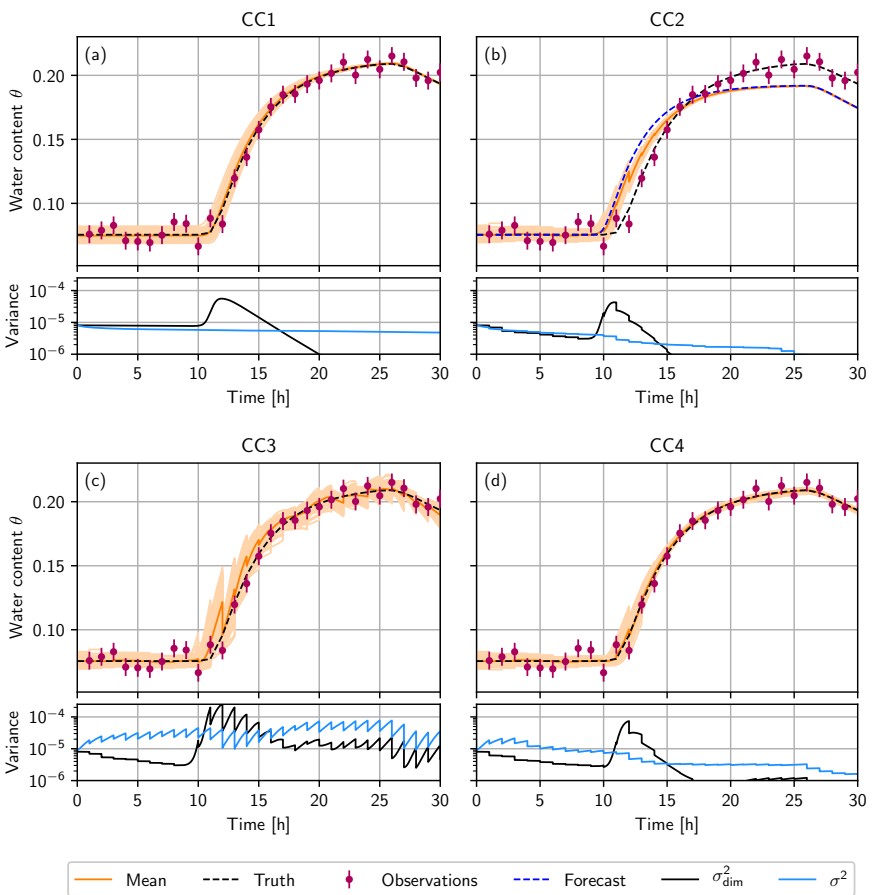

**Figure 2.** (a): Convergent case 1 (CC1): forward run without data assimilation, using an interpolated initial condition. In the other three cases the truth as used to generate the initial ensemble. (b): Convergent case 2 (CC2): the ensemble is propagated with $n = 2.68$ instead of $n_{\text{true}} = 2.28$ and the state is estimated. (c): Convergent case 3 (CC3): the parameter error is represented by the ensemble using $\mathcal{N}(2.68, 0.4^2)$ and the state is estimated. (d): Convergent case 4 (CC4): simultaneous state and parameter estimation. Top panels: The ensemble mean (orange) and the ensemble (light orange lines) during the forward run at the depth of the uppermost observation (20 cm). The truth, which is used to generate the observations (purple), is shown as a black dashed line. (b) additionally shows a single forward run (blue dashed line) using a wrong parameter $n = 2.68$ starting from the true initial condition. Bottom panels: Mean variance $\sigma^2$ (light blue line) of the ensemble over all dimensions and variance $\sigma^2_{\text{dim}}$ (black line) at the depth of 20 cm.

This case shows that the predictability for a perfect convergent system is infinite. After a transient phase, the states converge to the truth (Kalnay, 2003). A perfect model is not what we encounter in reality, however.





### 4.2.2 Convergent case 2 (CC2) – state estimation, wrong parameter

In this case, the state is estimated with the EnKF but the ensemble is propagated with a wrong parameter. Instead of $n_{\text{true}} = 2.28$,

$n = 2.68$ is chosen. The mean for the initial state is chosen as the true initial water content.

    In Fig. 2b (top panel) the temporal development of the water content in the depth of the uppermost observation (20 cm) is shown. A larger $n$ results in an earlier arrival of the infiltration front at the depth of 20 cm for the ensemble than for the truth. The EnKF tries to correct the ensemble but fails because its variance is too small and cannot represent the truth. Due to the convergent system, $\sigma^2$ decreases constantly while $\sigma^2_{\text{dim}}$ decreases fast to zero after the infiltration front reaches the depth of

20 cm (see Fig. 2b, bottom panel). This convergence leads to a false trust in the model and the filter degenerates. Compared to a forward run without data assimilation, the EnKF can only improve the state estimation for a short time when the water content rises due to the infiltration front. Soon after, the ensemble coincides with the free forward run and the estimated state shows no advantage any more.

    This case illustrates that a wrong parameter in a convergent system can lead to filter degeneration. This is in direct contrast

to DC2 (Fig. 1b), where the filter is still able to estimate the state. The behavior also differs from DC3 (Fig. 1c), where the observation interval in the Lorenz-96 model is too short such that it cannot develop its full divergent behavior. There the filter can also degenerate for a wrong parameter, but the data assimilation is still able to improve the estimation compared to a free forward run, since the ensemble never colapses entirely.

### 4.2.3 Convergent case 3 (CC3) – state estimation, represented error

In this case, the parameter error is represented with the ensemble but the parameter is not estimated. Each ensemble member has a different parameter $n$. The parameters are Gaussian distributed with $\mathcal{N}(2.68, 0.4^2)$ such that the truth lies within one standard deviation. The mean for the initial state is chosen as the true initial water content.

    Figure 2c (top panel) shows the temporal development of the water content in a depth of 20 cm. The infiltration front reaches this depth at different times due to the different parameter $n$ for each ensemble member. This increases the variance in the

ensemble both, in this depth and overall (see Fig. 2c, bottom panel). The variance increases rapidly between the observations, similar to the divergent cases. This way, the ensemble spread stays large enough such that the EnKF can correct the states. The ensemble can follow and represent the truth. This behaviour can also be observed for the divergent case DC4 with a short observation interval (Fig. 1d).

    Representing the model error adds a divergent component to the ensemble for a convergent model. This allows the EnKF

to correct the state and follow the truth. However, the predictability of the system decreases since each ensemble member converges to a different fixed point apart from the truth. To increase the predictability, parameter estimation is necessary.





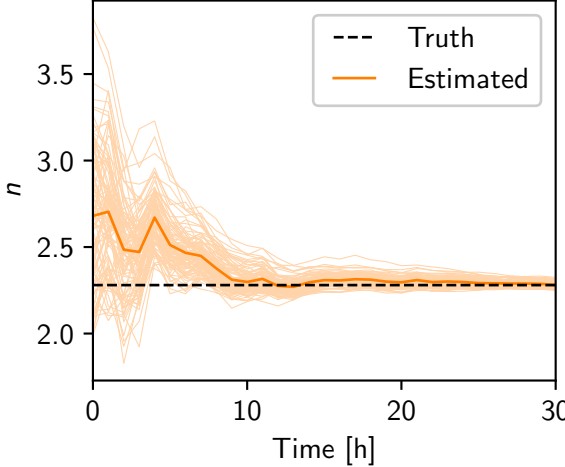

**Figure 3.** Estimation of parameter $n$ in convergent case 4 (CC4). The ensemble mean is shown in orange and the ensemble in light orange. The truth is a black dashed line.

### 4.2.4 Convergent case 4 (CC4) – state and parameter estimation

In this case, the error in the parameter $n$ is not only represented but also estimated as part of the augmented state. The initial parameter set is, as in CC3, Gaussian distributed with $\mathcal{N}(2.68, 0.4^2)$ such that the truth is located within one standard deviation.

Again, the mean for the initial state is chosen as the true initial water content.

The estimation of $n$ is shown in Fig. 3. The ensemble converges to the truth in a fast way because only one parameter is estimated, so every deviation from the truth is mainly caused by this parameter.

The mean variance $\sigma^2$ increases initially (Fig. 2d, bottom panel), because in the beginning the parameter has not been sufficiently improved such that the ensemble members still have different $n$. This leads to a divergent ensemble in state space

during the infiltration similar to CC3. While the parameter is estimated, the variance of the ensemble decreases fast and the convergent property of the system becomes dominant.

The temporal development of the water content in a depth of $20\,\mathrm{cm}$ is shown in Fig. 2d (top panel). In contrast to CC3, the corrections of the EnKF to the state are much smaller. The mean of the parameter $n$ comes close to the true value and the uncertainty of $n$ decreases. This causes the forward propagation to come close to the true model as well. The propagation with

an almost correct model supports the state estimation due to the convergent nature of the system which forces the state to the true value.





## 5 Discussion

For the divergent Lorenz-96 system, the EnKF is able to estimate the state for the true model as well as for the case with a wrong parameter. In a divergent system, the volume of the prior in phase space increases during forward propagation (Evensen, 1994). For the EnKF, this is directly connected to the ensemble spread, which increases rapidly between the observations. This prevents a collapse of the ensemble even in the presence of an unrepresented parameter error. However, if the observation interval is too small such that the system cannot develop its full divergent behavior, the EnKF leads to a decrease in the ensemble spread such that the filter degenerates in the case of an unrepresented parameter error. Nevertheless, the divergent behavior of the model prevents a complete collapse of the ensemble, so that the filter is still able to improve the state in a limited way.

In the convergent soil hydrological system, the volume of the prior distribution decreases during forward propagation such that the prior becomes more certain during the forward propagation even without an observation or data assimilation. For a perfect model, the predictability and state estimation in a convergent model are trivial. The initial ensemble will converge to the truth after some time, even with a rough approximation. In this case, data assimilation is not necessary.

In the case of model errors, in our study realized through a wrong parameter, the situation is different. The ensemble converges to a wrong state, the filter degenerates and data assimilation fails. Increasing the ensemble size can only improve the performance marginally since all ensemble members converge to the same fixed point.

Representing the parameter error by assigning each ensemble member a different parameter, increases the divergence of the system and the filter is able to estimate the state again. Between the observations, the ensemble spread increases rapidly because the ensemble members diverge to different fixed points apart from the truth. This results in a finite predictability. By representing the parameter error, the Richards equation behaves more similar to the divergent Lorenz-96 model. In the case of a convergent system, it is necessary to represent the parameter error, otherwise the ensemble collapses.

To increase the predictability of the system again, it is necessary to not only represent but also reduce uncertainties in the parameter. In synthetic cases without model structural errors, the convergent property of the system supports the state estimation and the predictability increases, if the parameter estimation is successful. This shows the importance of parameter estimation for convergent systems. For divergent systems parameter estimation also increases the predictability but only up to a point, because predictability is limited by the system's dynamics.

For the application of data assimilation to real data, model errors typically cannot be attributed to unknown parameters alone, but also stem from model structural errors like a simplified representation of sub-scale physics or unrepresented processes in the dynamics. While divergent models can alleviate the effect of an unrepresented error within bounds, in a convergent it is necessary to represent all relevant model errors sufficiently to prevent filter degeneracy and enable an optimal state and parameter estimation. However, representing these model errors is challenging. For example in hydrology, the model errors are typically ill-known (Li and Ren, 2011) and can vary both in space and time, which then can lead to filter degeneracy and biased parameter estimation (Berg et al., 2019).





Alternatively, inflation methods, can help to increase the ensemble spread and avoid filter degeneracy. For example, keeping
a constant ensemble spread for the parameters can provide a sufficient spread in state space (Zhang et al., 2017). The advantage
is that this approach is a model-specific inflation and avoids overamplification of spurious correlations (Constantinescu et al.,
2007). The disadvantage is, that the EnKF is prevented from reducing the prediction uncertainty. This behavior is also shown
in CC3. In this case, the ensemble spread in the parameter space adds a divergent component to the ensemble that results in an
increased ensemble spread in state space and prevents filter degeneracy.

   Multiplicative inflation increases the ensemble spread through an inflation factor (Anderson and Anderson, 1999). In a
divergent system a small multiplicative inflation is sufficient to increase the existing ensemble spread. In a convergent system
this requires a larger inflation and can lead to overinflation of spurious correlations (Constantinescu et al., 2007). To avoid
this and to cope with spatial and temporal varying model errors, the use of more sophisticated adaptive inflation methods (e.g.
Bauser et al., 2018; Gharamti, 2018) may be necessary in convergent systems.

   Inflation methods cannot fully replace the representation of model errors and can lead to biases in the estimation of pa-
rameters. If unrepresented model errors can be identified and are limited in space or time, these biases can be prevented by
using a closed-eye period for the extent of the unrepresented model errors. In the closed-eye period, parameters are kept con-
stant and only the state is estimated, which can require inflation methods for a successful state estimation. This enables an
improved parameter estimation without compensating the unrepresented model dynamics though biased parameters outside of
the closed-eye period, when and where uncertainties can be represented. The use of the closed-eye period in combination with
the representation of relevant uncertainties has been demonstrated in hydrology (Bauser et al., 2016).

## 6    Conclusions

We demonstrated the difference and challenges of ensemble data assimilation for divergent and convergent systems on the
example of the EnKF applied to the divergent Lorenz-96 model and a convergent soil water movement model based on the
Richards equation.

   Sequential ensemble data assimilation methods require a sufficient divergent part in the ensemble to maintain an adequate
ensemble spread and prevent filter degeneration. In divergent systems this is inherent to the system. In convergent systems
relevant model errors must be represented to increase the ensemble spread and add a divergent part to the ensemble to avoid
filter degeneracy. If errors stem from unknown parameters, estimating the parameters improves state estimation. However, this
will reduce the ensemble spread again and require the remaining relevant model errors to be represented. Since this can be
challenging, increasing the model errors artificially, by limiting the reduction of parameter uncertainty or through inflation
methods, can be required in convergent systems.

   This paper highlights the challenges when transferring sequential ensemble data assimilation methods from divergent sys-
tems to convergent systems, which must be considered when applying data assimilation.



*Author contributions.* HHB and DB designed and implemented the presented study. DB performed and analyzed the simulations. All authors participated in continuous discussions. HHB and DB prepared the manuscript with contributions from KR. The presented results are based on the PhD thesis by DB.

*Competing interests.* The authors declare that they have no conflict of interest.

*Acknowledgements.* This research was funded by Deutsche Forschungsgemeinschaft (DFG) through project RO 1080/12-1. Daniel Berg was supported in part by the Heidelberg Graduate School of Mathematical and Computational Methods for the Sciences (HGS MathComp), funded by DFG grant GSC 220 in the German Universities Excellence Initiative. Hannes H. Bauser was supported in part by the Deutsche Forschungsgemeinschaft (DFG) through Project BA 6635/1-1.





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
