# Peer review of "Technical note: Sequential ensemble data assimilation in convergent and divergent systems"

_Hydrology and Earth System Sciences, 2020_

## Author Comment (AC1)

**Answer to Review by Referee #1**

*In this paper, the authors investigate the difference in the impact of data assimilation for convergent and divergent systems. They conclude that a different approach is needed for these two types of systems.*

*I have a dual opinion of the paper after having read it. On the one hand, this is an issue that has been under-investigated in hydrology, and could use some more research. On the other hand, there are a number of issues with the paper that really need to be addressed before the paper can be published.*

**Reply:** We thank the reviewer for the fast response and confirming the relevance of the work. We have revised the manuscript accordingly and attached a file showing the differences. We refer to this file when giving line numbers in our response.

Several of the raised specific comments address a common issue regarding the generation of an adequate ensemble. We first discuss this issue in general, before addressing the specific comments:

The reviewer argues that an adequate generation of the ensemble avoids filter degeneration and that consequently methods to counteract filter degeneracy, like inflation methods, are not necessary. If we neglect issues like spurious correlations, we agree with the reviewer. Then, if all model errors are represented, the ensemble is generated adequately. We also agree, that this is possible in a synthetic cases, where these errors can be represented. However, in real-world systems, this is extremely difficult or even impossible based on the available process understanding.

We would like to illustrate this on the example of soil water movement, which is used in this study as an example for a convergent system. We describe soil water movement using the Richards equation. The Richards equation is based on several assumptions when deriving this macroscopic continuum equation (from a pore scale description). One of these assumptions is the so called local equilibrium assumption, which assumes that the changes in the external forcing are slow compared to the time scales of the equilibration at the local scale. This assumption is often valid, but can be violated for example during strong rain events. If this assumption is violated, the employed equation itself is not valid anymore. An accurate representation of this error is beyond the currently available knowledge.

Of course, some uncertainties to cover this error could be introduced, for example through assigning different parameter values to different ensemble members. However, this is by far not an adequate ensemble. Not only will these errors be represented rather crudely when they occur and lead to wrong correlations to unobserved locations leading to physically wrong predictions in the analysis, but also will the model error be by far too large when the local equilibrium assumption holds and erroneously bias the estimation towards the measurements. Therefore, parameter uncertainties should only reflect the uncertainties in the parameter, not more. This shows that creating such an adequate ensemble can be extremely difficult, or even impossible.

Please also note, that the approach of representing uncertainties through model parameters has been called model-specific inflation in the literature (Constantinescu et al., 2007).

We show in the manuscript that in a divergent system, it may not be necessary to represent all the uncertainties fully in the model to generate an adequate ensemble, due the divergent dynamics, which increases model uncertainties additionally. In a convergent system it becomes more relevant to represent the errors. This means it becomes more difficult or impossible to generate such an adequate ensemble in real-world cases. We illustrated this by not representing a parameter error in both systems. While such a parameter error can be represented well, model structural errors (such as the local equilibrium assumption in the Richards equation) can't.

If it is not possible to generate an adequate ensemble, this means, a strict approach that relies on the generation of an adequate ensemble is not practicable, particularly in convergent systems. This then shows that the use of heuristic methods to counteract filter degeneracy (i.e. inflation methods) becomes necessary.

We have improved this description throughout the manuscript in the response to the individual issues below.

*Introduction Line 32: In ensemble data assimilation methods, ..., this leads to decreasing uncertainties over time, which favors filter degeneration. If the filter degenerates, that means that the ensemble is not generated correctly. An adequate generation of the ensemble should prevent this.*

**Reply:** We agree, that an ensemble that accurately describes all model uncertainties could prevent this. As mentioned above this may not be possible. At this location in the introduction, our aim is to point out the difference between convergent and divergent systems. We altered the sentence slightly to clarify that our statement refers to a case when only state uncertainties are represented in the model. In this case our statement remains true.

We changed the statement to (lines 32-34):

"Several geophysical system, such as soil hydrology or atmospheric chemical transport are convergent systems. In ensemble data assimilation methods, if uncertainties are only represented through an ensemble of states, this leads to decreasing uncertainties over time, which favors filter degeneration."

*Line 37: You are forgetting uncertainties in the meteorological forcings!*

**Reply:** Meteorological forcings are included in the term 'boundary conditions'. To make this clearer, we extended the phrase to 'boundary conditions, which include external forcings'.

Lines 37-39 in the manuscript now read:

"The largest uncertainties in convergent systems typically do not reside in uncertain initial conditions but rather in boundary conditions, which include external forcings, the representation of sub-scale physics through parameterizations, and unrepresented physics in the model equations."

*Line 40: The Kalman filter has been developed to estimate states. In system theory, a parameter is defined as a variable that does not change. I know it is common practice to also update parameters, but it is worth noting that this violates the theoretical basis of the Kalman filter.*

**Reply:** We agree that the Kalman filter has been developed to estimate states. We also agree, that parameters are defined as variables that do not change. That is why the forward model for parameters is typically assumed to be constant to reflect this. The parameter estimation in the data assimilation is then used to reduce initial uncertainty of the parameter value. For example Evensen (2007) presented the theoretical basis for the joint estimation of states and parameters for the EnKF. We therefore decided to not apply any changes to the manuscript.

*Line 43: A random walk is also often used as parameter model.*

**Reply:** We have extended the section.

Lines 42-47 in the manuscript now read:

"Estimating parameters in ensemble data assimilation methods through an augmented state requires a forward model for the parameters as well. This model is typically assumed to be constant, which is neither divergent nor convergent. However, the filter will gradually reduce the uncertainty in the parameters, which is not increased through a divergent forward model and challenges similar to convergent systems can arise. This is sometimes alleviated by assigning a random walk as the forward model to the parameters, which then requires to determine an appropriate step size, however."

*Line 47: Inflation methods are a complete violation of the theoretical foundation of the Kalman filter. The update equations are derived by minimizing the posterior state error. Adding this inflation will, by default, lead to suboptimal results. A correct ensemble generation will solve the need to do this.*

**Reply:** As mentioned in our response above, it may not be possible to generate an appropriate ensemble.

Please note, that there is also a fundamental justification for increasing the ensemble spread, even if an appropriate ensemble can be generated: Ensemble methods necessarily have a finite number of ensemble members leading to spurious correlations. These will then lead to a too small ensemble spread (Houtekamer

and Mitchell, 1998). For this, van Leeuwen (1999) gives the theoretical justification. This means that the EnKF has a fundamental issue, due to a finite ensemble size, that leads to too small uncertainties. Ways to counteract this are needed. This will then lead to improved results. Therefore, we disagree. Inflation methods have the potential to improve the results if applied appropriately.

Nevertheless, this is an important point and inflation methods are heuristic. We added in the manuscript (Lines 49-51):

"This would require an adequate representation of all uncertainties, including unrepresented physics in the model equations. In real world systems this is often difficult or impossible. Therefore, practical alternatives are necessary, which are often heuristic and may interfere with basic assumptions in the data assimilation methods, however."

*Line 51: If the parameter uncertainty is kept constant, the Kalman filter is effectively reduced to optimal interpolation. This should be clarified.*
**Reply:** For the parameter part, keeping the uncertainty constant, indeed corresponds to a pre-defined covariance matrix as in optimal interpolation. However, if this is applied within an augmented state, the state component of the covariance matrix is not fixed and still calculated by propagating the ensemble with the model equations. Therefore, we prefer not to make this connection to optimal interpolation here.

*Line 53: Same comment as for line 47. Using this damping factor is a complete contradiction of the theoretical foundation of the Kalman filter. The author of that paper is Hendricks Franssen, by the way.*
**Reply:** Thank you for pointing out the error in the citation, we corrected it in the bibtex file. We added an explanation to the beginning of the paragraph in response to the comment on line 47 addressing the violations of theoretical foundations.

*Line 107-111: This is very unclear. Please think about reformulating in a manner to make this easier to digest.*
**Reply:** We simplified the description. We limited it to the the leading Lyapunov exponent, which is sufficient for this manuscript.

Lines 112-120 now read:

"The Lyapunov exponent quantifies how fast two initially infinitesimal close trajectories will separate. Analyzing the leading Lyapunov exponent for $F = 8$ shows a doubling time of $\tau_d = 0.42$ units for the distance between two initially infinitesimally close neighboring states (Lorenz and Emanuel, 1998). Increasing the forcing leads to a more divergent system, for instance $\tau_d = 0.3$ for $F = 10$ (Lorenz, 1996). With $\tau_d$ decreasing, so does the predictability of the system."

*Figure 1: Does "forecast" mean the open loop (forward) run?*
**Reply:** Yes, we replaced 'Forecast' with 'Forward run' to make this clearer.

*Line 150: Do you mean Fig. 1c?*
**Reply:** Yes, thank you for noticing. We corrected this in the manuscript.

*Line 189: Usually, in soil moisture data assimilation studies, only the surface layer soil moisture is assimilated. Please justify this strategy. Also, an observation error of 0.007 seems quite low.*
**Reply:** Assimilating just the surface layer may suffice for atmospheric applications. Ours is the common approach in vadose zone hydrology (e.g., Li and Ren, 2011; Wu and Margulis, 2013; Song et al., 2014; Erdal et al., 2014, 2015; Bauser et al., 2016; Brandhorst et al., 2017; Valdes-Abellan et al.,2019). The observation error of 0.007 is rather low. It corresponds to high quality measurements as obtained with time domain reflectometry (TDR)-sensors.

We extended the description in the manuscript to (Lines 195-197):
"Four time domain reflectometry (TDR)-like water content observations are generated equidistantly at depths of $(0.2, 0.4, 0.6, 0.8)$ m. The observation error is chosen to be $\sigma_{\mathrm{Obs}} = 0.007$ (e.g., Jaumann and Roth, 2017)."

*Line 192: This is very important. It is stated: To generate the initial ensemble, the ensemble mean is perturbed by a correlated multivariate Gaussian distribution. This is very unclear. Please provide a better explanation on how the ensemble is generated, as this is very important, and could help interpret the results later in the paper.*
**Reply:** The initial ensemble is generated by perturbing the initial water content state. The perturbation is spatially correlated to avoid an immediate spatial dissipation of the perturbations.

We extended the description in the manuscript to (Lines 198-205):
"To generate the initial ensemble, the ensemble mean of the water content state is perturbed by a correlated multivariate Gaussian distribution using a Cholesky decomposition to create an ensemble that corresponds to a predefined covariance matrix (e.g., Berg et al., 2019). The main diagonal of this covariance matrix of the ensemble is $0.003^2$. The off-diagonal entries are determined by multiplying the variance on the main diagonal with the the fifth-order piecewise rational function by Gaspari and Cohn (1999) using a length scale of $c = 10$ cm. This ensures a spatially correlated initial state, which increases the diversity of the ensemble. If instead uncorrelated Gaussian random numbers with a zero mean were used, the dissipative component of the system would lead to a fast dissipation of the perturbation in space for each individual ensemble member."

*Immediately below figure 2: This case shows that the predictability for a perfect convergent system is infinite. I do not agree. It shows that the ensemble is not adequately generated.*
**Reply:** In this case no data assimilation is employed. A perfect model is initialized with different initial states. Since this is a convergent system, these states converge to the truth, if a perfect model is employed (which includes perfect boundary conditions, etc.). The truth is predicted with an increasing accuracy and decreasing uncertainty. Of course, a perfect system will not occur (which is also stated in the manuscript). However, it is important to understand this fundamental difference when applying data assimilation to convergent systems.

To clarify this we extended the description in the manuscript to (Lines 220-222):
"This case shows that the predictability for a perfect convergent system is infinite (in contrast to divergent systems, that only have a finite limit of predictability even if the model including boundary conditions is perfectly known). After a transient phase, the states converge to the truth (Kalnay, 2003). A perfect model is not what we encounter in reality, however."

*Line 230: I would argue that disturbing the n values leads to an ensemble that is well generated, which explains the better results.*
**Reply:** We agree. In this synthetic case the model error is completely known and consequently can be represented in the ensemble. We clarify this in the manuscript. Please also note, that this approach has been called a model-specific inflation method (Constantinescu et al., 2007).

We extended the description in the manuscripts (Lines 240-243)
"In this case, the parameter error is represented with the ensemble but the parameter is not estimated. Each ensemble member has a different parameter $n$. The parameters are Gaussian distributed with $\mathcal{N}(2.68, 0.4^2)$ such that the truth lies within one standard deviation. Since the model error is known in this synthetic case we can create an ensemble that represents the model error adequately. Note, that this is more difficult or even impossible in a real-world system."

*Line 268: A model is never perfect, so a better word than "perfect" should be used.*

**Reply:** We agree. Still, we can explore the consequences if the model were prefect.

*Line 283: Again, you are forgetting meteorological forcings.*
**Reply:** We added boundary conditions to this list.

Lines 295-297 now read:

"For the application of data assimilation to real data, model errors typically cannot be attributed to unknown parameters or uncertainties in boundary conditions alone, but also stem from model structural errors like a simplified representation of sub-scale physics or unrepresented processes in the dynamics."

*Line 290, 291 and 296: Again, inflation, and using a constant ensemble spread, are tricks that are used to compensate for a poorly generated ensemble.*
**Reply:** To better motivate the relevance of inflation methods we have improved the prior paragraph and have changed the introduction to this paragraph that covers inflation methods.

Lines 295-307 now read:

"For the application of data assimilation to real data, model errors typically cannot be attributed to unknown parameters or uncertainties in boundary conditions alone, but also stem from model structural errors like a simplified representation of sub-scale physics or unrepresented processes in the dynamics. Uncertainties in parameters and boundary conditions can be represented in an ensemble, but representing model structural errors is challenging or can be impossible. For example, in hydrology, the model errors are typically ill-known (Li and Ren, 2011) and can vary both in space and time, which then can lead to filter degeneracy and biased parameter estimation (Berg et al., 2019). While divergent models can alleviate the effect of an unrepresented error within bounds, in a convergent system it is necessary to represent all relevant model errors sufficiently to prevent filter degeneracy and enable an optimal state and parameter estimation. Since this is challenging or even impossible, heuristic ways to address filter degeneracy are necessary.

A practical alternative to increase the ensemble spread and avoid filter degeneracy are inflation methods. [...]"

We have extended our criticism on inflation methods in Lines 317-318:

"Heuristic inflation methods cannot fully replace the representation of model errors and hence must be used judiciously. They can lead to overinflation of spurious correlations and can lead to biases in the estimation of parameters."

*As stated above, I am in favor of the idea of the study, as this is important. But the issues I mentioned should be resolved before the paper can be published.*
**Reply:** We thank the reviewer for raising the issues, which helped clarifying the manuscript.

**Additional References**

Brandhorst, N., Erdal, D., and Neuweiler, I.: Soil moisture prediction with the ensemble Kalman filter: Handling uncertainty of soil hydraulic parameters, *Advances in Water Resources*, 110, 360–370, https://doi.org/10.1016/j.advwatres.2017.10.022, 2017.

Evensen, G., *Data Assimilation: The Ensemble Kalman Filter*. New York: Springer, 2007.

Erdal, D., Neuweiler, I., and Wollschläger, U.: Using a bias aware EnKF to account for unresolved structure in an unsaturated zone model, *Water Resources Research*, 50, 132–147, https://doi.org/10.1002/2012WR013443, 2014.

Erdal, D., Rahman, M., and Neuweiler, I.: The importance of state transformations when using the ensemble Kalman filter for unsaturated flow modeling: Dealing with strong nonlinearities, *Advances in Water Resources*, 86, 354–365, https://doi.org/10.1016/j.advwatres.2015.09.008, 2015.

Houtekamer, P. L., and H. L. Mitchell, A sequential ensemble Kalman filter for atmospheric data assimilation, *Monthly Weather Review, 129*(1), 123–137, doi:10.1175/1520-0493(2001)129¡0123:ASEKFF¿2.0.CO;2, 2001.

Li, C. and Ren, L.: Estimation of unsaturated soil hydraulic parameters using the ensemble Kalman filter, *Vadose Zone Journal*, 10, 1205– 1227, https://doi.org/10.2136/vzj2010.0159, 2011.

Song, X., Shi, L., Ye, M., Yang, J., and Navon, I. M.: Numerical comparison of iterative ensemble Kalman filters for unsaturated flow inverse modeling, *Vadose Zone Journal*, 13, https://doi.org/10.2136/vzj2013.05.0083, 2014.

Valdes-Abellan, J., Pachepsky, Y., Martinez, G., and Pla, C.: How Critical Is the Assimilation Frequency of Water Content Measurements for Obtaining Soil Hydraulic Parameters with Data Assimilation?, *Vadose Zone Journal* 18, doi:10.2136/vzj2018.07.0142, 2019

van Leeuwen, P. J., Comment on "Data assimilation using an ensemble Kalman filter technique", *Monthly Weather Review, 127*(6), 1374–1377, doi:10.1175/1520- 0493(1999)127¡1374:CODAUA¿2.0.CO;2, 1999.

Wu, C.-C. and Margulis, S. A.: Real-time soil mois- ture and salinity profile estimation using assimilation of embedded sensor datastreams, *Vadose Zone Journal*, 12, https://doi.org/10.2136/vzj2011.0176, 2013.

[revised manuscript text omitted]

---

## Author Comment (AC2)

**Answer to Review by Referee #2**

***General comments:*** *The technical note titled "Sequential ensemble data assimilation in convergent and divergent systems" written by Hannes Helmut Bauser et al. investigated the performance of data assimilation in convergent and divergent systems. Although the manuscript showed how a sequential data assimilation method (EnKF) works in two convergent and divergent systems, at the current version, manuscript only showed the results of different case studies without detailed explanation why we observed those results, which is crucial for technical studies. In addition, the manuscripts did not provide suggestions to improve the performance of data assimilation in convergent and divergent systems. In my opinion, it is better if the manuscript considers scenarios that use one or two techniques to maintain ensemble spread or to avoid convergence to wrong states in convergent systems as stated in the introduction section. Examples presented in the manuscript is ideal. Authors may consider examples that are nearer to the realistic cases.*
**Reply:** We thank the reviewer for the comments and recognizing our work. We have revised the manuscript accordingly and attached a file showing the differences. We refer to this file when giving line numbers in our response.

The aim of this technical note is to show the fundamental difference between convergent and divergent systems in ensemble data assimilation. In our understanding a technical note focuses on a specific constrained aspect, and is not an extensive assessment. We therefore chose two extreme cases in this study, that allow to illustrate these differences, but these are simplified compared to real-world applications. Nevertheless, they highlight the key aspects that also have to be considered in the real-world applications. The different systems themselves and their individual performance with data assimilation have been studied in the literature and is not part of this note.

This note focuses on creating understanding of the fundamental behavior and differences. Solutions to improve the performance of data assimilation will be problem specific. This can also be seen in the wide range of different inflation methods (one possible heuristic solution that have limitations) that are applied. Again, the application of such methods has been studied for the individual systems. Specific results found may not be transferable to a different system.

We have extended our description to clarify the choice of the illustrative cases. We have also improved explanations throughout the manuscript (see also responses to specific comments below).

Lines 69-75: "We investigate and demonstrate the different challenges, that illustrate for example the different requirements for inflation methods, using the ensemble Kalman filter (Evensen, 1994; Burgers et al., 1998). The divergent case is illustrated using the Lorenz-96 model (Lorenz, 1996), while for the convergent case, a soil hydrological system described by Richards' equation is used. Naturally, these are highly simplified models compared to real-world applications. Still, they demonstrate key aspects that also have to be addressed in more complicated situations. The specific adjustments applied there depend on the particular model. Our focus here is on the fundamentals, not on the wide range of specifics. "

Lines 106-107: "This section demonstrates the data assimilation behavior for a divergent system on the example of the 40-dimensional Lorenz-96 model, which has been widely used to test data assimilation methods in atmospheric sciences (e.g., Li et al., 2009)."

***Specific comments:***

*- Introduction: Evidences on why we need to assess the performance of data assimilation in convergent and divergent system should be more clearly stated (why we need to do this study). How convergent/divergent system impacts on the effectiveness of data assimilation? What is the disadvantage of divergent systems (slowly converge to the truth state?) ? Provide more examples of convergent/divergent systems in hydrology and geosciences.*

**Reply:** In the data assimilation literature geophysical systems are typically not distinguished between convergent and divergent. To better introduce this differentiation we have included a new Fig. 1 to show the qualitative differences between the two systems. In the introduction, the motivation for this study is then grounded on several indicators that the differentiation between convergent and divergent systems may yield interesting insights. (i) data assimilation is applied to convergent and divergent systems (ii) the development of data assimilation methods is mainly driven by the atmospheric sciences (iii) the observation that in convergent systems sometimes rather dramatic choices are made that ensure a sufficient ensemble spread (e.g. keep parameter uncertainty constant). The goal of this paper is then to provide basic insights into this difference and emphasize the need and bring additional attention to the challenges from model structural errors in data assimilation in convergent systems (which is a finding of this paper and rather a fundamental challenge to be addressed by future research).

To make this motivation clearer in the introduction we made several changes:

Lines 21-22 and Fig. 1: "In this study we distinguish geophysical systems between divergent and convergent systems, depending on the development of two initially nearby states (Fig. 1)."

Lines 26-27: "However, this is only true for perfect models, which is usually not the case for geophysical systems. This can lead to a bias with convergence to a wrong state."

Lines 32-35: "While weather forecasting or oceanography are divergent systems, several geophysical system, such as soil hydrology or chemical transport are convergent systems. In ensemble data assimilation methods, if uncertainties are only represented through an ensemble of states, this leads to decreasing uncertainties over time, which favors filter degeneration."

Lines 37-38: "The application of data assimilation when coupling a divergent and a convergent model, for example in coupled chemistry meteorology models, may lead to potentially new difficulties (Bocquet et al., 2015)."

Lines 62-64: "Similarly, Constantinescu et al. (2007) showed that an atmospheric chemical transport model required much stronger inflation than reported in the meteorological literature and showed better results for a model specific inflation, where the key parameters are perturbed to achieve an increased spread in the state."

Lines 69-71: "We investigate and demonstrate the different challenges, that illustrate for example the different requirements for inflation methods, using the ensemble Kalman filter (Evensen, 1994; Burgers et al., 1998)."

[Figure]

Figure 1: Dynamics of a generic divergent and convergent dynamic system with different initial states. Both panels show a single state dimension of a multi-dimensional system. In the divergent system, initially infinitesimally close states drift apart, while in the convergent system initially apart states converge.

*- Section 3 and 4: Provide more technical explanation of the obtained results and show the quantitative criteria (NSE, relative error, correlation coefficient, etc.) that compare results obtained with data assimilation and with forecast (open loop), especially explain the changes of ensemble variance with time. Why the initial condition for data assimilation was not set to be different from the true one? What is the observation values used for assimilation and their corresponding errors?*

**Reply:** We have extended the explanations for the observed results in sections 3 and 4. The goal of this manuscript is to provide insight and understanding on the differences between convergent and divergent systems with regard to data assimilation. Therefore, we focus on the qualitative features in the data assimilation. By deliberately not providing summarizing statistics, we guide the reader to actually look at the figures to understand the behavior.

We provide the RMSE for the reviewer. The RMSE for the divergent cases of the analysis state to the synthetic truth in the shown state dimension at measurement times are DC1: 0.3, DC2: 0.7, DC3: 2.2, DC4: 0.4. Comparing the free forward run (with a wrong parameter, which is the same for DC2-DC4) to the synthetic truth at the 8 measurement times as in DC2, leads to an RMSE of 6.8, comparing truth and forward run at the 80 measurement times as in DC3 and DC4 yields an RMSE of 5.4. The numbers confirm the qualitative observations. The RMSE for the convergent cases of the analysis state to the synthetic truth in the shown state dimension at measurement times are CC1: 0.001, CC2: 0.012, CC3: 0.003, and CC4: 0.001. Comparing the free forward run (with a wrong parameter, which is the same for CC2-CC4) to the synthetic

truth at the 30 measurement times yields an RMSE of 0.014. Again, this agrees with the qualitative observations.

The variances are shown in the lower panel for each case in Figures 2 and 3 (former Figures 1 and 2). The overall behavior is that the variance increases during the forecast for the divergent system, while it decreases for the convergent system. The analysis step reduces the variance at the measurement times.

To illustrate the behavior of the divergent system, it was not necessary to start from a biased initial condition. This ensures to keep the chosen examples more simple and understandable.

We have made the following changes to extend the explanations:

Line 94: "By combining the information from measurement and model, the uncertainty in the analysis ensemble is decreased."

Lines 130-132: "The behavior of the EnKF on a divergent system is investigated through four different cases (DC1-4), which are purposely designed simple to illustrate the behavior concisely. For all cases, the model is solved using a fourth order Runge-Kutta method with a time step of $\Delta t = 0.01$."

Lines 145-146: "Uncertainty only stems from the uncertainty in the initial condition."

Lines 148-149: "Due to the divergent nature of the model, the ensemble spread increases between observations and the ensemble has a sufficient spread such that the EnKF is able to correct the states to follow the truth."

Lines 156-157: "In this case, in order to investigate the impact of an unrepresented model error, the ensemble is propagated with a wrong parameter of $F = 10$ instead of $F = 8$ (Fig. 2b), which was used to generate the observations."

Lines 165-167: "The high frequency of analysis steps, where the Kalman update reduces the variance in the ensemble, prevents the Lorenz-96 model to develop enough divergence to increase the ensemble spread sufficiently in the short time intervals in between (see Fig. 2c, bottom panel)."

Lines 177-178: "This increase is sufficient to help the filter such that it does not degenerate (Fig. 2d, top panel). Representing the parameter error in the ensemble in the case of frequent observations can prevent filter degeneracy."

Lines 223-224: "Due to the convergent behavior of the model in combination with the true representation, the initially broad ensemble converges to the truth, even though the initial condition was not represented accurately."

Lines 342-344: "Sequential ensemble data assimilation methods require a sufficient divergent part in the ensemble to maintain an adequate ensemble spread and prevent filter degeneration. In divergent systems this is inherent to the system, provided that observation intervals and divergence times match."

In regard to the observation values and errors, the model settings for the synthetic truth are given in subsections 3.2 and 4.2. The details for observations and their errors are already stated in lines 138-143: "Synthetic observations are generated in all 40 dimensions by a forward run until time 4, using the true value and perturbing it with

a Gaussian distribution with a zero mean and a standard deviation of $\sigma_{\mathrm{obs}} = 1.0$. For cases DC1 and DC2 observations are generated at 8 different times with an observation interval of $\Delta t_{\mathrm{Obs}} = 0.5$. This observation interval is chosen rather large to ensure a large divergence of the system. For cases DC3 and DC4 observations are generated at 80 different times with an observation interval of $\Delta t_{\mathrm{Obs}} = 0.05$. This interval is often used in other studies (e.g. Nakano et al., 2007; van Leeuwen, 2010; Poterjoy, 2016)." and in lines 206-208: "Four time domain reflectometry (TDR)-like water content observations are generated equidistantly at depths of $(0.2, 0.4, 0.6, 0.8)\,\mathrm{m}$. The observation error is chosen to be $\sigma_{\mathrm{Obs}} = 0.007$ (e.g., Jaumann and Roth,2017). Observations are taken hourly for a duration of $30\,\mathrm{h}$."

*- Section 4: Which method was used for joint state and parameter estimation?*
**Reply:** We clarified this and specify 'state augmentation method' instead of 'as part of the augmented state'.

Lines 268-269 now read:
"In this case, the error in the parameter $n$ is not only represented but also estimated using the state augmentation method."

*- Figure 1: Show the results corresponding with forecast in figures 1a,b,d for comparison. Explain what is and .*
**Reply:** We have added the forward run, starting from the truth but with wrong parameter (F=10) to Figures 2b and 2d as well. We did not add it to figure 2a, where the truth as well as the ensemble is propagated with the same parameter (F=8). Hence, a comparison with a free forward run with F=10 is not needed and a free forward run with F=8 but again starting from the true initial state would just correspond to the synthetic truth again.

In regards to the request to 'Explain what is and .', it seems like the symbols are not displayed in the comment. We did find in the figure caption that we did not specify that $x_2$ corresponds to the state dimension 2 and have added this in the caption.

The corresponding part of the figure caption now reads:
"Top panels : The ensemble mean (orange line) and the ensemble (light orange lines) in the data assimilation run for state dimension 2 $x_2$ together with the observations (purple), generated from the truth (black dashed line). (b-d) additionally show a single forward run (blue dashed line) using a wrong parameter $F = 10$ starting from the true initial condition."

*- Figure 2: results obtained with the forward run without data assimilation should present in figures b,c,d for comparison.*
**Reply:** We added the forward run to Figures 3c and 3d and adjusted the caption accordingly. Figure 3b already contains the forward run. We did not add it to figure 3a, since in this cases ensemble and and truth use the same parameter.

The corresponding part of the figure caption now reads:

[revised manuscript text omitted]